# Metadata Standard for Continuous Preservation, Discovery, and Reuse of Research Data in Repositories by Higher Education Institutions: A Systematic Review

Neema Florence Mosha * and Patrick Ngulube

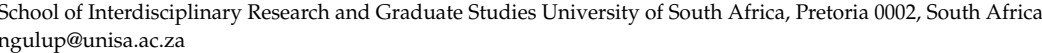

School of Interdisciplinary Research and Graduate Studies University of South Africa, Pretoria 0002, South Africa; ngulup@unisa.ac.za
* Correspondence: moshanf@unisa.ac.za

**Abstract:** This systematic review synthesised existing research papers that explore the available metadata standards to enable researchers to preserve, discover, and reuse research data in repositories. This review provides a broad overview of certain aspects that must be taken into consideration when creating and assessing metadata standards to enhance research data preservation discoverability and reusability strategies. Research papers on metadata standards, research data preservation, discovery and reuse, and repositories published between January 2003 and April 2023 were reviewed from a total of five databases. The review retrieved 1597 papers, and 13 papers were selected in this review. We revealed 13 research articles that explained the creation and application of metadata standards to enhance preservation, discovery, and reuse of research data in repositories. Among them, eight presented the three main types of metadata, descriptive, structural, and administrative, to enable the preservation of research data in data repositories. We noted limited evidence on how these metadata standards can be used to enhance the discovery and reuse of research data in repositories to enable the preservation, discovery, and reuse of research data in repositories. No reviews indicated specific higher education institutions employing metadata standards for the research data created by their researchers. Repository designs and a lack of expertise and technology know-how were among the challenges identified from the reviewed papers. The review has the potential to influence professional practice and decision-making by stakeholders, including researchers, students, librarians, information communication technologists, data managers, private and public organisations, intermediaries, research institutions, and non-profit organizations.

**Keywords:** developing countries; higher education institutions; metadata; metadata standards; research data repositories; researchers

## 1. Introduction

Research data preservation, discovery, and reuse have been a central part of most of the mission and vision of higher education institutions (HEIs) and their entities, such as universities, libraries, archives, and repositories [1] with many other stakeholders involved, including researchers, funders, students, administrators, technicians, and service providers. The main aim is to ensure that research data that are created daily are well preserved, discovered, and reused, particularly in the contemporary Open Science (OS) landscape [2]. The framework provided by the findable, accessible, interoperable, and reusable (FAIR) [3]; collective benefit, authority to control, responsibility, and ethics (CARE); and transparency, responsibility, user focus, sustainability, and technology (TRUST) [4,5] principles enable the long-term preservation and reusability of research data in digital repositories [2,5]. The established methods to preserve, describe, and reuse research data using metadata and metadata standards [5–7].

Metadata are data about a digital resource that is stored in an organised form suitable for machine processing, and they attend to many purposes in long-term preservation,

including providing a record of activities that have been performed upon the digital material and supporting the discovery and reuse of research data [8–10]. A metadata standard is a requirement that is intended to establish a common understanding of the meaning of the data to ensure proper use and interpretation of data by its creator and other users [6,11]. There are different forms of metadata standards, including structure, value, content, and interchange standards [12]. The scope of metadata standards includes ontologies, taxonomies, name authority files, and other types of knowledge organisation systems (KOS) [7]. The National Information Standards Organization (NISO) divided metadata into three main groups: descriptive metadata, administrative metadata, and structural metadata [13–17]. Administrative metadata are further classified into technical metadata, right metadata, and preservation metadata [14,18].

Metadata also guarantee the preservation of a digital resource/object, for example, archived sites and through specific metadata standards, such as Preservation Metadata Implementation Strategies (PREMIS) and Metadata Encoding and Transmission Standard (METS) [6,19]. Other metadata standards include Dublin Core, Encoded Archival Description (EAD), Visual Resources Association (VRA) Core, Categories for the Description of Works of Art (CDWA), and Machine-Readable Cataloging (MARC) and Metadata Object Description Schema (MODS) [6,19,20]. The Resource Description Access (RDA) metadata standard catalogue is a collaborative, open directory of metadata standards applicable to research data [21]. Several metadata standards have been identified and evaluated to fit into required research data and datasets into repositories [5,6]. Metadata are an indispensable component of any research data repository's definition, purpose, and function because it is the basis of the practical creation and maintenance of metadata standards that enable the management of research data [22,23]. If metadata records are formatted to a common standard, they can facilitate the readability of the metadata by both humans and machines and machine to machine [24,25]. Thus, to make research data publicly accessible and reusable, researchers need to deposit their raw data and datasets into repositories and provide metadata records that conform to the repository's metadata schema [9,26–29]. A metadata schema is an entity, including the semantic components and content (called a set of metadata elements), such as encoding the metadata with a syntax or markup language like the Machine-Readable Cataloguing (MARC) format and an eXtensible Markup Language (XML)/Standard Generalised Markup Language (SGML) DTD, which has three basic parts or characteristics [24,30]:

(a) Structure—data model or architecture used to hold the metadata and the way the metadata statements are expressed. As examples, we can mention the RDF metadata architecture and the XML METS schema.

(b) Semantics—names and meanings of the elements and their refinements.

(c) Contents—statements or instructions of how and what values should be assigned to the elements.

Using a standardised metadata schema improves data interoperability and allows diverse datasets to be merged or aggregated in meaningful ways [10]. Research data loss in most repositories starts with the wrong or a lack of metadata standards. Thus, the use of metadata standards embedded in digital data from the outset is recommended as a starting point for research data preservation [26,31]. Whether the loss occurs by a malicious attempt or an inadvertent mistake, it can be diminishing either personally, machine, or to the institute/company where it occurs. To be useful, metadata need to be standardised [26,32]. This includes agreeing on language, spelling, date format, etc. [32]. If no metadata standards are used, it can be very difficult to archive, cite, discover, reuse, etc. research data and datasets researchers to create every day [32]. Many HEIs have supported initiatives to formalise the metadata specifications the community deems to be required for data reuse [22,26,32–34]. Therefore, this systematic review analysed available and open metadata standards to enhance the preservation, discovery, and reuse of research data in repositories in HEIs. Specifically, this review:

(a) Identified metadata types and their importance on research data preservation in

(b)    HEIs.

(c)    Ascertained the creation and application of metadata standards in HEIs.

(d)    Examined the available and online metadata standards to support the preservation, discovery, and reuse of research data in HEIs repositories.

(e)    Mentioned challenges impeding the use of metadata standards and providing potential solutions.

This review focused on the four areas because of various reasons. Metadata types are very important for preserving research data in repositories. The three types of metadata, which are descriptive, structural, and administrative, providing important information such as title, subject, author(s), format, size, keywords, and copyright, enhance preserving research data in repositories facilitating the discovery and reuse of such data for more research projects. Furthermore, elements such as the author, subject-specific, search, title, language, date of publication, subject covering, and call number are co-applied with a highly structured and searchable resource [11].

The creation and application of metadata standards because HEIs were also chosen because it explains the purpose and requirements for using metadata standards on research data produced by HEIs. It also provides the need for HEIs to involve various stakeholders in the creation and application of metadata standards. Researchers and HEIs can promote the preservation, discovery, and reuse of research data in the HEIs' repositories by using the openly available online metadata standards. Using these freely available metadata standards allows HEIs to identify, document, and share them with researchers with the assistance of various stakeholders, such as librarians and ICTs staff within HEIs, even though some may require training and knowledge on using them. Discussing the challenges preventing the use of metadata standards and recommending potential solutions could help HEIs work through the various issues brought up by the reviewed papers and implement any potential solutions to improve the use of metadata standards that HEIs will use to help researchers preserve, find, and reuse research data in repositories.

## 2. Methods

### 2.1. Data Sources and Search Strategy

This study reviewed existing research articles in the English language published in academic journals between 2003 and 2023. The dates are determined by the adoption of the open research data paradigm, in which most researchers store their research data in open data repositories to facilitate public sharing and reuse of such data [35,36]. The following databases served as data sources: Emerald Insight, Web of Science (WoS), ScienceDirect, and Scopus. Google Scholar (GS) was used to augment the results as recommended by Halevi, Moed, and Bar-Ilan [37]. GS is a web search engine that may gather trustworthy material that occasionally may not be found in other academic or scientific databases [38]. The search strategy was performed with the help of Boolean operators. The search strategy comprised the following keywords "Metadata standard AND Research communities AND Research data AND Preservation AND Reuse AND Repositories".

### 2.2. Screening

Researchers searched and screened the research articles by titles and abstracts for inclusion. Furthermore, researchers conducted the full-text screening. Rayyan® online software was used for screening. We adopted the following inclusion and exclusion criteria [39]:

(a)    Relevance for the review question.

(b)    Date of publication: 2003 to 2023.

(c)    Geographical location: Higher education institutions.

(d)    Types of publication: Research articles.

(e)    Nature of research: Systematic review.

(f)    Language: English.

### 2.3. Inclusion and Exclusion Criteria

The following inclusion and exclusion criteria guided this review, as shown in Table 1.

**Table 1.** Inclusion and exclusion criteria.

| Inclusion Criteria | Exclusion Criteria |
|---|---|
| Articles published in the English Language | Articles that were not published in the English Language |
| Articles that were published between the year 2003 and 2023 | Articles that were published before the year 2003 and after the year 2023 |
| Literature with substantial focus on metadata, metadata standards, research data, research data preservation, research data discovery, research data reuse, repositories. | Literature which did not focus on metadata, metadata standards, research data, research data preservation, research data discovery, research data reuse, repositories. |
| Peer reviewed research articles | Systematic reviews, scoping reviews, meta-analysis, rapid reviews, government and non-governmental organisation reports and academic dissertations and theses, editorials, book reviews, unpublished manuscripts, and conference abstracts. |
| Survey, qualitative (documentary, semi-structured interview, observation, case studies), and use cases. | Case-control, randomised control trials. |

### 2.4. Data Extraction and Coding

Data extraction was carried out under the guidance of the Preferred Reporting Items for Systematic Reviews and Meta-Analyses (PRISMA) checklist [40]. Researchers independently extracted data from the included studies using a standardised data extraction form using an open-access online tool (CADIMA) developed through a collaboration between the Julius Kühn-Institut and the Collaboration for Environmental Evidence [41] to increase the efficiency of the evidence synthesis process and facilitate reporting of all activities to maximise methodological rigour [41]. Afterwards, the data were compared. On the other hand, the data coding was done manually. Data coding conducted in systematic reviews indicates information, such as author(s), study design, date, and findings [42]. Table 2 illustrates data extraction and coding as conducted in this review.

**Table 2.** Data extraction and coding.

| Author(s) | Title | Journal | Study Design | Findings |
|---|---|---|---|---|
| Anil Hirwade [13] | A study of metadata standards | *Library Hi Tech News* | A survey was developed to examine the use, planning and evaluation of metadata standards. | Twenty metadata standards, that are OAI compliant, were studied including METS and MODS indicated general metadata standards, and learning object metadata (LOM) as educational materials and learning objects. |
| Chapepa, Ngwira, and Mapulanga [33] | | *Digital Library Perspectives* | Qualitative approach (interview and documentary review) with a case study strategy that focuses on the in-depth holistic and in-context examination of one or more cases | Dublin Core was selected as the only metadata standard to create and implement metadata |
| Christianson et al. [14] | A metadata reporting framework (FRAMES) for synthesis of ecohydrological observations | *Ecological Informatics* | Scientist-centred design, observation and interview with data originators and data consumers | The study developed a framework for reporting data and metadata for earth systems. FRAMES utilises best practices for data and metadata organization enabling consistent data reporting and compatibility with a variety of standardised data protocols. |

**Table 2.** *Cont.*

| Author(s) | Title | Journal | Study Design | Findings |
|---|---|---|---|---|
| Christianson et al. [14] | A metadata reporting framework (FRAMES) for synthesis of ecohydrological observations | *Ecological Informatics* | Scientist-centred design, observation and interview with data originators and data consumers | The study developed a framework for reporting data and metadata for earth systems. FRAMES utilises best practices for data and metadata organization enabling consistent data reporting and compatibility with a variety of standardised data protocols. |
| Donaldson, Zegler-Poleska, and Yarmey [15] | Data managers' perspectives on OAIS designated communities and the FAIR principles: mediation, tools, and conceptual models. | *Journal of Documentation* | Semi-structured interview | The use of the open archival information system (OAIS) reference model (ISO-14721) for the internal preservation of data. |
| Mayernik and Liapich [43] | The role of metadata and vocabulary standards in enabling scientific data interoperability: A study of earth system science data facilities. | *Journal of eScience Librarianship* | Case study to examine the consistency of metadata schema and subject vocabulary use within specific communities. | ISO 19115:2003 and DataCite metadata standards are used by more than 40% of the data facilities and repositories. |
| Kim et al. [44] | Comprehensive knowledge archive network harvester improvement for efficient open-data collection and management. | *Electronics, Telecommunications, and Information (ETRI) Journal* | Observing and investigating the functionalities of the Comprehensive Knowledge Archive Network (CKAN), an open-source data distribution platform. | The study derives the problems of CKAN in terms of data inconsistency and storage space waste for data deletion. Based on these observations, the study proposed an improved CKAN that provide a new deletion function solving data inconsistency. |
| Klöcking et al. [45] | Community recommendations for geochemical data, services, and analytical capabilities in the 21st century. | *Geochimica et Cosmochimica Actan.* | Case study | The Ecological Metadata Language (EML) is an XML-based metadata specification developed for the description of datasets and their associated context in ecology. The conversion of EML metadata to an ontological form has been addressed in existing observation ontologies, which are able of providing a degree of computational semantics to the description of the datasets, including the reuse of scientific ontologies to express the observed entities and their characteristics |
| Mena-Garcés et al. [46] | Moving from dataset metadata to semantics in ecological research: a case in translating EML to OWL | *Procedia Computer Science* | Observation | The Ecological Metadata Language (EML) is an XML-based metadata specification developed for the description of datasets and their associated context in ecology. The conversion of EML metadata to an ontological form has been addressed in existing observation ontologies, which are able of providing a degree of computational semantics to the description of the datasets, including the reuse of scientific ontologies to express the observed entities and their characteristics. |

**Table 2.** *Cont.*

| Author(s) | Title | Journal | Study Design | Findings |
|---|---|---|---|---|
| Formenton and de Souza-Gracioso [6] | Metadata standards in web archiving technological resources of archived websites. | *Digital Journal of Library and Information Science* | qualitative, exploratory, and descriptive research was done, using the bibliographic method from a non-systematic inventory together with a review and analysis of the literature content. The Dublin Core, MODS, EAD, visual resources association (VRA) core, PREMIS, and METS standards were selected and analysed. | Dublin Core, MODS, EAD, and VRA Core supported METS and PREMIS in detecting and documenting technical aspects of sites and proving their authenticity, context, and origin. METS can manage archived sites by acting as OAIS information packages, while Dublin Core proved to be an exponent for Web archiving through its use in remarkable area initiatives. |
| Wu et al. [18] | Metadata creation practices in digital repositories and collections: schemata, selection criteria, and interoperability. | *Data Intelligence* | A survey on which metadata schema has been adopted by participating data repositories and presents an analysis of crosswalks from fourteen research data schemas to Schema.org. | Most descriptive metadata are interoperable among the schemas, the most inconsistent mapping is the rights metadata, and a large gap exists in the structural metadata and controlled vocabularies to specify various property values. |
| Park and Tosaka [9] | Metadata creation practices in digital repositories and collections: schemata, selection criteria, and interoperability. | *Information Technology and Libraries* | The study examines the prevailing current state of metadata-creation practices in digital repositories, collections, and libraries, which may include both digitised and born-digital resources. | MARC, AACR2, and LCSH are the most widely used metadata schema, content standard, and subject-controlled vocabulary, respectively. Dublin Core is the second most widely used metadata schema, followed by EAD, MODS, VRA core, and TEI. Qualified Dublin Core's wider use vis-à-vis Unqualified Dublin Core (40.6 percent versus 25.4 percent) is noteworthy. Existing technological infrastructure and staff expertise also are significant factors contributing to the current use of metadata schemata and controlled vocabularies for subject access across distributed digital repositories and collections. |
| Dietrich [46] | Metadata management in a data staging repository. | *Journal of Library Metadata* | The study reviews DataStaR project by presenting high-level use cases. It follows with a description of DataStaR's metadata architecture, focusing on the semantic Web components that facilitate metadata reuse and the creation of metadata according to multiple standards. | DataStaR employs a semantic metadata management architecture that provides several key benefits to users and librarians. It uses a Web-based interface to create metadata and export valid XML in multiple standards, the ability to reuse previously created metadata in a straightforward manner, and compatibility with emerging semantic Web technologies. |

**Table 2.** *Cont.*

| Author(s) | Title | Journal | Study Design | Findings |
|---|---|---|---|---|
| Burke et al. [32] | Using existing metadata standards and tools for a digital language archive: A balancing act. | *The Electronic Library* | Use cases whereas it discusses some of the areas important for representing language materials where both University of North Texas Libraries (UNTL) metadata and CoRSAL metadata practices were adapted to better fit the needs of intended audiences. | All records in the UNT Libraries' Digital Collections use a uniform metadata scheme (UNTL) based on the Dublin Core standard with added local fields and qualifiers for more specificity and greater flexibility. UNTL has 21 fields including eight that are required: main title, language, content description, subject (2), resource type, format, collection, and institution. |

## 3. Findings of the Search Results

The search retrieved 1597 research papers, and 967 non-English language and duplicated research papers were excluded. A total of 630 research papers were selected for further screening, and 351 research papers were excluded following title screening. A total of 276 potential research papers were selected, and 198 research papers were excluded following the abstract screening. An amount of 81 relevant research papers were selected and retrieved, and 58 research papers were excluded following full-text screening. A total of 23 research papers were subjected to quality assessment, 10 research papers were excluded, and 13 potential research papers were used in this review. The selection of the retrieved research papers for this review is illustrated in Figure 1. Table 3 presents the main findings of this review.

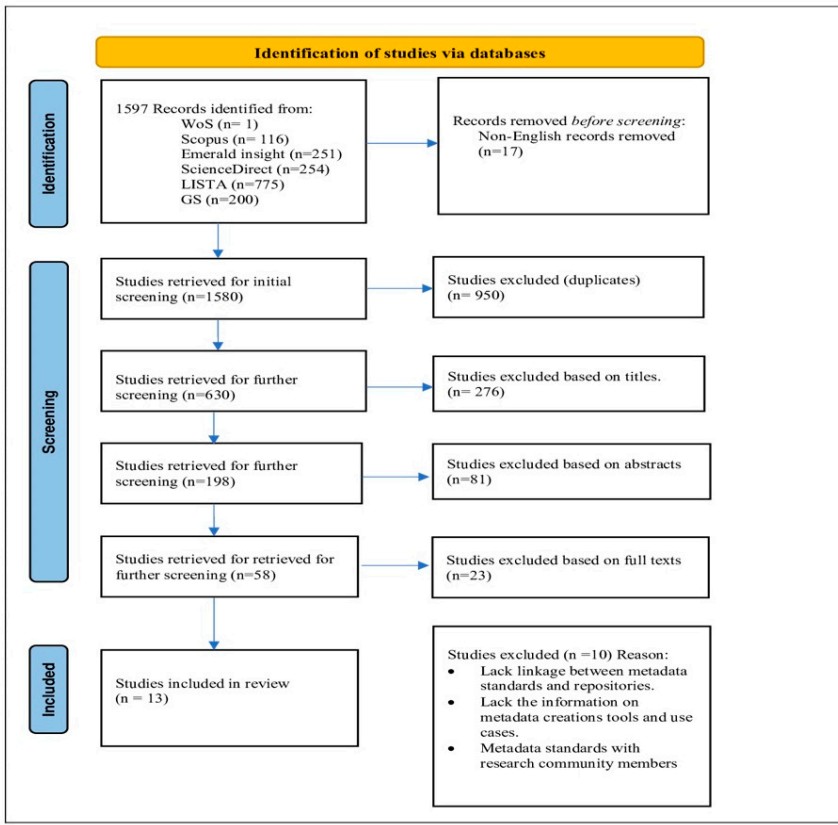

**Figure 1.** PRISMA flow diagram, metadata standards, research data, preservations, repositories.

**Table 3.** Main findings.

| Themes | Research Article(s) | Main Findings |
|---|---|---|
| Metadata types and their importance to research data preservation in HEIs | [14,18] | Application of metadata and metadata standards on research data preservation, discovery, and reuse. |
| | [15,43] | OAIS reference model has been widely adopted as essential in digital preservation and it provides a technical architecture for data repositories. |
| | [14] | Repositories or data centres may guide or even dictate the content and format of metadata used for preserving data, using a formal metadata standard. |
| | [6,13,14,18] | Three main types of metadata: descriptive, structural, and administrative (administrative—technical, right and preservation) that provide information such as title, author(s), abstract, extent, subject, publisher, keywords, data collection and analysing tools, page numbers, format, date, and the geographic location of the data in the repository, facilitates data discovery, organises electronic resources, promotes interoperability, and supports the curation and preservation of research data in the repository. |
| The creation and application of metadata standards in HEIs | [6,13,14,46] | Involvement of stakeholders such as researchers, students, citizens, librarians, and information and communication technologists (ICT) in HEIs to enhance the creation and development of metadata and metadata standards for their research community members to preserve, discover, and re-use research data in repositories. |
| | [9,33,45] | Metadata standards can be created by humans (manually) or machines (automatically) using recommended guidelines and applications. However, manual creation requires knowledge of metadata structure schemes, content standards, and controlled vocabulary schemes and repositories designs which can comply with metadata standards for research data to be preserved. |
| Available metadata standards to enhance the preservation, discovery, and reuse of research data in repositories. | [14,18,46] | Openly available metadata standards are available to make research data more valuable for HEIs by making it more discoverable, reusable, and preservable. |
| Available metadata standards to enhance the preservation, discovery, and reuse of research data in repositories. | [14,18,46] | Openly available metadata standards are available to make research data more valuable for HEIs by making it more discoverable, reusable, and preservable. |
| | [6,18] | There are general metadata standards used to describe almost any data, and specific metadata standards that meet the unique requirements of certain industries, domains, and disciplines.<br>There are also generic metadata standards that are widely adopted and easy to use, and domain specific metadata standards that are specialised and richer in vocabulary to be used in a specific discipline. |
| | [43,44] | Metadata standards grow out of a community need or through a formal standardisation body, such as W3C—data catalogue vocabulary (DCAT), International Organization for Standards (ISO), International Federation of Library Associations and Institutions (IFLA), and the internet engineering task force (IETF). |

**Table 3.** *Cont.*

| Themes | Research Article(s) | Main Findings |
|---|---|---|
| Challenges impeding the use of metadata standards and investigate potential solutions. <br><br> (a)  Challenges | [9,14,28] | Lack of metadata and metadata standards policy and guidelines, lack of strategic partnerships, and lack of management support. |
| (b)  Potential solutions | [9,33,45] | ○  Make data more Findable, Accessible, Interoperable, and Reusable by both humans and machines (FAIR). <br> ○  Use updated and agreed metadata standards. <br> ○  Develop new metadata standards where required. <br> ○  Apply collaborative approach, in which a group of libraries and HEIs develop and share tools, skills, and services. <br> ○  Discuss and present the needs and requirements with HEIs' management for support such as funds, tools, and expertise for the creation and implementation of effective metadata standards and repositories. <br> ○  Conduct a suitable and regular training for technical teams, researchers, and HEIs. <br> ○  Establish a working relationship among librarians, data depositors, and repository administrators. |

### 3.1. Metadata Types and Their Importance on Research Data Preservation in HEIs

Reviewed papers emphasised that research data must be accompanied by metadata standards to ensure that they are adequately preserved to allow users to discover and reuse them [6,14,15,18]. The current review noted that over the years, HEIs communities have been developing metadata standards that define how to reliably preserve research data in repositories [14,15,18]. For example, since 2002, OAIS reference model (ISO-14721) has been widely adopted as essential in digital preservation [14]. The metadata standard establishes a common way of structuring and understanding data and includes principles and implementation issues for utilising the standard provided [18]. Repositories or data centres may guide or even dictate the content and format of metadata used for preserving data using a formal metadata standard [14]. Review papers [14,18] outlined the three main types of metadata and their significance on research data preservation as follows.

### 3.2. Descriptive Metadata

Descriptive metadata facilitate data discovery within both data centres and data repositories [6], and they enhance the search and retrieval of the content [14]. Descriptive metadata provide essential information about the data, such as title, author(s), abstract, extent, subject, publisher, keywords, data collection and analysing tools, and the geographic location of the data in the repository [18]. Descriptive metadata organise electronic resources, promote interoperability, and support the curation and preservation of research data [6,32].

### 3.3. Structural Metadata

Structural metadata provide unique identifiers, page numbers, and special features (tables of contents, indexes) [6,32]. They also provide a relationship between two datasets and general relation between two datasets [18] and support the linking among the components of a resource [18].

### 3.4. Administrative Metadata

Administrative metadata manage digital objects and provide information about the ownership, file type, size, compression format, date of creation, access permissions (open or closed), preservation event, copyright status, and license terms where primary uses

are interoperability, digital object management, and preservation [6,18]. Administrative metadata are subdivided into:

(a) Technical metadata indicate the technical aspects and dependencies of a digital file to decode and render it [6,18], and describe the rules, structure, and format for storing data example data models, data lineage, and backup rules [6].

(b) Rights metadata provide information about the rights held in and over the resource, whereas the license is a sub-property of the rights, which is defined as the legal document giving official permission to do something with the resource [18].

(c) Preservation metadata contain information required for the long-term management of digital data and the migration to other digital formats as software and hardware change continuously [14,18]. Preservation description metadata are necessary for the long-term archiving of research data, such as provenance, checksums, and unique identifiers [14].

Additionally, the Dublin Core metadata element set is made up of 15 elements (title, creator, subject, description, publishers, contributors, date, type format, identifier, source, language, relation, coverage, and right) and addresses the various elements obtained among metadata types (descriptive, administrative, and technical), which are also needed to identify digitised resources and data [17,18,32]. In general, metadata types are critical to establishing an accurate understanding of the nature of resource items of:

(a) Content based on what an object contains or is about, such as subject headings;

(b) Context based on factors related to the creators of the object, such as authors, who, what, why, where, and how; and

(c) The data's organisational structure, including the chapters and articles that make up the data [18,19].

### 3.5. The Creation and Application of Metadata Standards in HEIs

The reviewed papers presented the need for metadata standards among HEIs to preserve as well as enhance the discovery and reuse of research data to a wider research community [6,18]. HEIs need to establish metadata creation tools to assist the creation of metadata standards that should integrate with the established workflows to encourage documentation of research data [47]. Over time, the authors have developed extensive guidelines providing usage information and example values to enhance the creation of metadata standards within organisations [32]. This includes general, system-wide guidelines at: https://library.unt.edu/digital-projects-unit/metadata/input-guidelines-descriptive/ (accessed on 5 April 2023) and collection-specific instructions at: https://library.unt.edu/digital-projects-unit/metadata/ (accessed on 5 April 2023) project-specific-guidelines-documents/ to provide support for editors of different skill levels [32]. (Burke et al., 2022). Creating and applying metadata standards is becoming a top priority for most of HEIs [8] due to the following reasons:

(a) To create a standard set of guidelines for information tagging.

(b) To guarantee uniformity in the application of metadata.

(c) To encourage resource sharing and application interoperability.

(d) To open the door for cutting-edge technology.

The current review also noted that metadata standards could be created by humans (manually) or machines (automatically) using recommended guidelines and applications [28]. However, manual creation requires knowledge of metadata structure schemes, content standards, and controlled vocabulary schemes and repository designs, which can comply with metadata standards for research data to be preserved [9,33,45]. The review noted the involvement of researchers, students, citizens, librarians, technical resources, and information and communication technologists (ICTs) within and outside HEIs [6,13,33] in the creation of metadata standards for their research data [32]. For example, an ecology researcher studies the spread of an invasive species and produces both spreadsheets and geographic information system (GIS) data [9,18]. Using the laboratory's

GIS software, the researcher creates the Federal Geographic Committee's Content Standard for Digital Geospatial Metadata (FGDC-CSDGM) metadata for the GIS data to prepare it for publication to a geospatial data repository with the help of library staff [47]. The DataStaR team has found that researchers are willing to create metadata, especially when it does not require a significant amount of extra effort [47]. Dietrich [47] adds that a librarian can help researchers using XML application and a crosswalk to transform the FGDC-CSDGM metadata standards to EML, which is required by the ecological data repository.

Two reviewed papers showed the support provided by HEIs to formalise the metadata standards for their research community members to enable them to preserve, discover, and re-use research data [13,14]. In other words, metadata standards emerged from the needs of research communities within HEIs [13,14]. Many different metadata standards are being developed as standards across disciplines, such as library science, education, archiving, medicine, e-commerce, and arts [13]. In general, the creation of metadata standards requires several stakeholders and agents, including resource creators, metadata experts, or library staff [32], and factors, such as resources; time, users; subject matter; staff expertise; repository system compatibility; interoperability; and budget [9].

### 3.6. Openly Available Metadata Standards to Support the Preservation, Discovery, and Reuse of Research Data in Repositories

The reviewed papers indicated the need for using openly available metadata standards to enhance research data to be more preservable, discoverable, and reusable in new research to add value [18,26]. Metadata still define the guarantee of preservation of a digital resource/object (for example, archived sites), through specific metadata standards, such as PREMIS and METS [6]. Metadata standards are classified into general and specific metadata standards, which deal with specific areas, such as health or transport [6,18]. Examples of general metadata standards are Darwin Core and MODS, which are used to describe almost any data [6], while the examples of specific metadata standards are the data document initiative (DDI) that meet the unique requirements of certain industries, domains, and disciplines [13,18]. Other metadata standards are classified into generic and domain-specific metadata standards [6]. Generic metadata standards are widely adopted and easy to use while domain-specific metadata standards are specialised and are richer in vocabulary to be understood by researchers in a specific discipline [8]. Rich sample and analytical metadata, such as provenance, description of method and analysis conditions, and the completeness of metadata, allow for the assessment of accuracy and precision and ensure reproducibility of research data [45]. Metadata standards are sometimes based on region or country [6].

In general, metadata standards grow out of a community need or through a formal standardisation body, such as W3C–data catalogue vocabulary (DCAT), International Organization for Standards (ISO), International Federation of Library Associations and Institutions (IFLA), and the internet engineering task force (IETF) [27,43]. Appropriate citation of the people, laboratories, organisations, HEIs, funders, and research artifacts are following appropriate metadata standards (e.g., the International Generic Sample Number (IGSN) for samples, the Open Researcher and Contributor Identifier (OR-CID) for authors, the Research Organisation Registry (ROR) for institutions, or the DataCite metadata standard [28]. DCAT is a resource description framework (RDF) that helps create a standardised way of setting up datasets in terms of descriptions (metadata) [18,43,44]. DCAT and Dublin Core include research data properties that are common to almost all types of datasets [18]. DCAT and DataCite metadata standards are used for general research data [43], (Mayernik and Liapich, 2022) while the European Clinical Research Infrastructure Network (ECRIN) schema is an extension of DataCite and GigaDB from the life sciences and biomedical domain, which are used to export metadata in general purpose metadata standards, such as DataCite and Schema.org [43]. The comprehensive knowledge archive network (CKAN) is an open-source data distribution platform for open data [44]. CKAN can be used with extensions, such as Datastore and Datapusher, for data management and harvesters and

DCAT for data collection [44]. The harvesters are configured to work with DCAT schema and DCAT fields to enhance the process [18,44]. Many datasets are on a CKAN portal, and the ability to push or pull datasets from one CKAN instance to another is extremely useful [18,44]. Reviewed papers presented metadata standards that are open archives initiative (OAI) compliant, including DC, AACR 2, METS, MODS, and LOM [13]. METS can manage archived sites by acting as OAIS information package, and it is used as an XML schema for encoding descriptive, administrative, and structural metadata regarding objects and data in a digital repository [6]. Most of the metadata standards from reviewed papers are discipline specific, such as EML, ISO 19115, Darwin Core, and DataStaR. EML was developed in ecology to consolidate various formats of ecological research data [13]. DDI and Text Encoding Initiative (TEI) provide representation of textual objects in humanities, social sciences, and linguistics information and data [13]. EAD is used to describe archives data, corporate records, and personal papers [13]. Mena-Garcés et al. [46] add that the EML is an XML-based metadata specification developed for the description of datasets and their associated context in ecology. The conversion of EML metadata to an ontological form has been addressed in existing observation ontologies, which are able to provide a degree of computational semantics to the description of the datasets, including the reuse of scientific ontologies to express the observed entities and their characteristics [46].

The ISO 19115 standard is designed for geographical data and geospatial community, and it is used to describe data quality, access, and right to use [18], and the information on the spatial and temporal schema, spatial reference, and distribution of digital geographic data [45]. Darwin Core is a metadata specification for information about the geographical occurrence of species and biological specimens [45–48]. DataStaR is a data staging repository currently in development at Cornell University Library designed to support the curation of scientific research data [47]. DataStaR provides a repository space with user-controlled access permissions that allow researchers to share datasets to publish datasets and related metadata to various external repositories [47,48], and it contains a set of web-based tools to create metadata in a variety of formats [47].

### 3.7. Challenges Impeding the Use of Metadata Standards and Provide Potential Solutions

(i)　　Challenges

The reviewed papers presented the following challenges that hinder the creation and application of metadata standards in research data repositories in HEIs:

### 3.8. Lack of Metadata and Metadata Standard Policy and Guidelines

HEIs and their libraries lack metadata standards' policies and guidelines, and most which are available policies do not define their metadata best practices and guidelines, most of the repositories do not have a well-defined metadata policy [9,33]. There is still a widespread lack of adoption of these policies by the research community to guide different activities, such as data sharing, including the additional effort of organising and formatting data, distrust and protection, copyright and licensing, and knowledge about the most appropriate repository [45].

### 3.9. Lack of National Web Archiving

Given the lack of national Web archiving studies that investigate, systematise, and analyse in depth the metadata and the characteristics of the metadata standards applicable in the preservation of digital data [6].

### 3.10. Vocabulary as Situated in Language

Consistency issues and a lack of adherence to established vocabularies or schemes restrict the ability of metadata to be related to other types of data [15]. Languages are complex communication systems; at the foundational level every language has many ambiguities, idiosyncrasies, inconsistencies, and nuances, which need the application of grammatical rules [45,48]. Common challenges occur when deciding on the name (label)

of a metadata property due to synonymous and homonymous terms, singular or plural word forms, lexical and dialectical variants, and an array of word forms (e.g., hyphenated, compound, or bound concept) [18,45].

### 3.11. Lack of Expertise and Resources

Most of HEIs lack detailed metadata standards and technical skills as well as required resources [25,45]. Furthermore, many digital initiatives lack adequate skills and resources [49]. Repositories development is a labour-intensive process, which needs knowledge and awareness among team members [18,25,45].

### 3.12. Repositories Designs

Repository designs in some of the repositories limit the accommodation of metadata standards for research data to be preserved [17]. Additionally, repositories are not funded and maintained; these problems arise because many of the data systems catering to a specific domain were born out of the research projects that succeeded in attracting funding to further develop their infrastructure [45].

### 3.13. Lack of Strategic Partnerships

Most libraries are unwilling to form strong collaborative relationships with those who possess the technical skills and expertise required to implement and maintain complex metadata tools and standards [45].

(ii)   Potential solutions

This review noted the following potential solutions for presented challenges that hinder the creation and application of metadata standards to research data in repositories in HEIs:

(a)   Research data should be accompanied by a unique PID, such as the handle system [45].
(b)   Develop new metadata standards where required [45].
(c)   Apply a collaborative approach between libraries and HEIs to enhance the development and share tools, skills, and services [9,33].
(d)   Discuss and present the need and requirements with HEIs' management for support such as funds, tools and expertise for the creation and implementation of effective metadata standards and repositories [33].
(e)   Conduct suitable and regular training for technical teams, researchers and HEIs [9,33].
(f)   Establish and improve the working relationship between ICTs and repository administrators [33].

## 4. Discussion

This review identified the need for metadata standards to enhance the preservation, discovery, and reuse of research data in repositories that metadata standards play a vital role in both FAIR and CARE guiding principles [3,5], whereas out of the 15 FAIR principles, 13 explicitly refer to metadata and metadata standards [3]. HEIs and researchers need to ensure that research data ranging from open data to shared research data can be freely used, reused, and shared by anyone for any purpose [3] by humans and/or machines [25]. This review noted the three main types of metadata: descriptive, administrative, and structural metadata. White [50] presented the use of descriptive metadata for data depositing in repositories. We noted other types of metadata and their importance, including business metadata, which describe business definitions, rules, and context for data, for example, data quality rules, report annotations, and glossaries [51]; operational metadata, which provide information on how and when data are created, for example, locations and data owners [52]; and usage metadata, which are used to provide information about how data are or have been used, for example, user rating, access-pattern metadata, and comments [53].

The review noted the involvement of stakeholders is one of the initiatives to ensure metadata standards created and used meet the need of the research community and their

environment. Reviews identified the need and requirements of each stakeholder. For example, for HEIs, human and technical resources are needed for the creation of metadata in academic repositories. Williams, Shankar, and Eschenfelder [16] indicated the need for more institution actors, such as scholarly disciplines, related professions, and the institutions' top members. Development of DDI was finalised through inter-organisational collaboration connected, where stakeholders worked together to create the DDI metadata standards [16]; they also established and maintained their specific orientations toward both the project and the boundaries between institutions, which led to the creation and successful implementation of metadata standards [16]. Chapepa, Ngwira, and Mapulanga [33] add that Qualified Dublin Core (QDC) was chosen by all participants as the only metadata structure scheme that they will use to create and implement metadata in the repository. Wierling et al. [54] add that metadata supported transparency of energy transition processes based on the validity of the basis for decision-making and collaboration across disciplines and societal groups to enhance the creation of metadata standards, there is a need to establish communication with users.

Reviews noted the following metadata standards that facilitate the preservation of research data in repositories: DCAT, DataCite, CKAN, METS, TEI, EAD, EML, ISO 19115, Darwin Core, and DataStarR. Other studies provided the following metadata standards: NISO MIX is a Z39.87 data dictionary technical metadata for digital still images (MIX), NISO metadata are for images in XML schema required to manage digital image collection [55], ISO/IEC 11179 is used to describe metadata and activities needed to manage data elements in a registry [55], market data definition language (MDDL) is used to map market data into a common language and structure to ease the interchange and processing of multiple complex data and datasets [56], and Security Assertion Markup Language (SAML) is an XML-based open standard data format for exchanging authentication data between parties [57].

Various challenges have been identified associated with metadata and metadata standard creation and usage in HEIs environment. Among the challenges noted in this review were the lack of metadata policy, the repositories' designs, and National Web archiving. Harvey, McLean, and Rzepa [58] also noted that repository design is among the challenges that affect the application of metadata standards among research data. Other studies reported challenges, such as a loss of granularity and inability to recreate the original metadata records and a lack of metadata standards that support the provision of a traditional field-based advanced search reflective of the granularity of the original records [59]; some metadata standards are built by consultants and employees that are not qualified, thereby creating problems in usage, lack of funds for implementation, maintenance, and poor selection of metadata standards [60]; Wrong metadata standards associated with research data in repositories can lead to difficulties in identifying and using the preserved research data [61], a lack of awareness and skills in creating and using metadata standards [62], a lack of management support that thus limits the ongoing procedures to be in place [63], and a lack of guidelines in describing resources, non-qualified or inexperienced metadata specialists, and research data depositors needed [59,62]. Training and awareness among stakeholders as well as the application of frameworks and models, such as DSP-PROV, which can be used to keep metadata schemas consistent over time can minimise the challenges identified [22]. Other potential solutions presented were monitoring and updating metadata standards and appointing people, organisations, data, software, instruments, and other research objects as per HEIs' mission, vision, and needs [25].

## 5. Conclusions

The review provided metadata types and their importance in preserving research data in repositories. The review showed a growing number of openly available online metadata standards to support research data preservation, discovery, and reuse. The review discussed three types of metadata that are descriptive, structural, and administrative and their importance on preserving research data in repositories. Most of the papers that

were reviewed discussed various kinds of metadata standards that are utilised in HEIs for research data preservation, discovery, and reuse; however, none of them mentioned how these standards were applied to research data in a particular HEI. The creation and application of metadata standards in HEIs were also presented and discussed in this review. The involvement of stakeholders, such as researchers, students, community members, and libraries, is deemed important for creating and using metadata standards in HEIs. Reviewed papers presented a need to incorporate more stakeholders in creating metadata standards for research data to enhance preservation, discovery, and reuse. However, they did not provide a clear picture of how these metadata standards enable the preservation, discovery, and reuse of research data in repositories. Most of the reviewed papers indicated the practical way of using metadata standards and not the theoretical part, which can provide more knowledge, especially for researchers in HEIs. Even though metadata standards are also meant for research data, assigning such metadata standards to research data needs experienced personnel to work with researchers; thus, more training and capacity building are needed not only for researchers, but also for more stakeholders engaging in metadata standards creation and management. For future studies, the incorporation of other research methods, such as follow-up telephone surveys and focus groups, is necessary to gain a fuller understanding of metadata standards and their usability to enhance research data preservation, discovery, and reuse in HEIs as well as the application of grey literature— specifically, handbooks and guidelines at specific schools.

**Author Contributions:** All authors contributed to conceptualizing and designing the study. All authors have read and agreed to the published version of the manuscript.

**Funding:** This research received no external funding.

**Data Availability Statement:** No new data were created.

**Conflicts of Interest:** The authors declare no conflict of interest.

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
