# Peer review of "Metadata Standard for Continuous Preservation, Discovery, and Reuse of Research Data in Repositories by Higher Education Institutions: A Systematic Review"

_information, doi:10.3390/info14080427_

Round 1

Reviewer 1 Report

Review to “Metadata standard for continuous preservation, discovery, and reuse of research data in repositories by higher education institutions: A systematic review

I enjoyed reading the manuscript entitled, “Metadata standard for continuous preservation, discovery, and reuse of research data in repositories by higher education institutions: A systematic review”.  The topic is timely and important to the field. However, I believe there are areas where the authors could make improvements to enhance the paper. 

·      Literature review: this paper aims to address metadata standard in four specific areas, as indicated on p. 3. To provide more clarity to the readers, it would be beneficial if the authors could offer a clear justification for why these four areas were chosen as the focus of the study. 

·      Methodology: Information in the methodology part is not adequate. 

*What types of studies were included in this systematic review? Can you explicitly state that in your inclusion/exclusion criteria?  

*It would also be valuable to mention whether journal papers, conference papers, or dissertations were included in the study, as this information is crucial for a systematic review. 

*I suggest that the authors provide reasons for exclusion not only in the final stage, but also during the full text screening stage if that is possible. 

*What methodology were the authors using to do the coding? How did the authors arrive at their results from the included studies? Currently, this process is not clear in the manuscript. 

·      Appropriateness of results and discussions: The results and discussion are good. However, it is necessary for the authors to elaborate on the methods employed to derive these results. By providing a clear explanation of the approach used, the readers will have a better understanding of how the authors arrived at their conclusions. 

·      Contribution of article to the profession: The overall topic of the study is undoubtedly valuable to the field. It addresses a pressing need and has the potential to make a significant contribution. 

I hope these comments are constructive and helpful for further revising the manuscript as you move forward. Wish you success!

Author Response

I responded to all the comments as per reviewer. The file is attached. 

Reviewer 2 Report

This seems to be a well-constructed review of the literature on metadata used in data repositories. I have to say that this is important! I've uploaded things to my own repository and no metadata was requested, which seems at odds with what we know (and you've found), so I suspect that it's a widespread problem, even in research extensive libraries. Here are some things that I recommend:

(1) Create a table with all of the articles that you included with main findings/emphases. Insert this after Figure 1.

(2) Summarize more in the conclusion – it also needs editing. For instance, “discipline specific based. None of the reviewed paper” (two grammatical errors there). It says “most of the reviewed papers” over and over; combine those, or make bullet points, maybe.

Instead of follow-up interviews for a future study, I would suggest also looking at the gray literature; specifically, handbooks and guidelines at specific schools. 

Author Response

I responded to all comments presented by reviewers. All the responses attached. 

Round 2

Reviewer 1 Report

The authors addressed the issues raised.